# Successful Treatment of Pediatric Inflammatory Multisystem Syndrome Temporally Associated with COVID-19 (PIMS-TS) with Split Doses of Immunoglobulin G and Estimation of PIMS-TS Incidence in a County District in Southern Germany

**DOI:** 10.3390/healthcare9040481

**Published:** 2021-04-18

**Authors:** Götz Wehl, Jörg Franke, Martin Frühwirth, Michael Edlinger, Markus Rauchenzauner

**Affiliations:** 1Department of Pediatrics, Hospital Ostallgäu-Kaufbeuren, 87600 Kaufbeuren, Germany; kinderklinik@kliniken-oal-kf.de (J.F.); markus.rauchenzauner@i-med.ac.at (M.R.); 2Department of Pediatrics, Hospital St. Vinzenz, 6511 Zams, Austria; office@krankenhaus-zams.at; 3Department of Medical Statistics, Informatics, and Health Economics, Medical University Innsbruck, 6020 Innsbruck, Austria; Michael.edlinger@i-med.ac.at; 4Department of Pediatrics, Medical University Innsbruck, 6020 Innsbruck, Austria

**Keywords:** PIMS-TS, COVID-19, adolescent, immunoglobulin G, myocarditis, PIMS-TS incidence

## Abstract

Pediatric inflammatory multisystem syndrome temporally associated with SARS Cov2 (PIMS-TS) is a newly encountered disease in children sharing clinical features with Kawasaki disease, toxic shock syndrome, or macrophage-activating syndrome. Pathogenically, it is associated with immune-mediated post-infectious hyperinflammation leading to short-term myocardial injury with yet unknown long-term outcome. We herein present three cases of PIMS-TS treated in our institution with divided doses of immunoglobulins and high dose acetyl salicylic acid, according to existing Kawasaki disease guidelines. Due to greater weight in adolescents affected and concerns of rheological sequelae following possible hyperviscosity, doses of immunoglobulins were divided and given 24 h apart with good tolerability. All patients recovered rapidly with normalization of previously encountered cardiac manifestations. As diagnosis of PIMS-TS should be made promptly, timing of therapy is of paramount importance for a favorable outcome. To date, no randomized controlled trial data exist concerning treatment recommendations. 1.8% (95% CI: 1.7% to 2.0%) of all children and adolescents in the county district of Ostallgäu were tested positive for SARS CoV-2, incidence of PIMS-TS was 1.7% (95% CI: 0.9% to 3.1%) among SARS CoV-2 positive tested earlier. As the pandemic is still ongoing, rising numbers of PIMS-TS in children might be expected.

## 1. Introduction

The COVID-19 pandemic declared in 2020 is a serious health threat across the entire world and mainly affects elderly adults, whereas disease burden in children and adolescents is reported to be milder, with only a few deaths reported due to COVID-19 [1]. Recently, case reports describing a Kawasaki-like disease with persistent fever, inflammation, and evidence of single or multi-organ dysfunction (shock, cardiac, respiratory, renal, gastrointestinal, or neurological disorder) in children and adolescents have been published [1,2,3,4,5]. This syndrome was referred to as multisystem inflammatory syndrome in children (MIS-C) associated with COVID-19 by the Centers for Disease Control and Prevention and as pediatric inflammatory multisystem syndrome temporally associated with SARS CoV-2 (PIMS-TS) in Europe [2], stating clinical and laboratory similarities between PIMS-TS and Kawasaki disease [6]. Six main elements of PIMS-TS are paramount and are accounted to across the different definitions: Pediatric age, persistence of fever, elevated laboratory markers of inflammation, manifestation of signs or symptoms of organ dysfunction, lacking an alternative diagnosis, and temporal relation to COVID-19 infection or exposure. In contrast to Kawasaki syndrome, PIMS-TS affects mainly older children and adolescents. To date, coronary aneurysms as a complication of PIMS-TS were rarely described but affection of myocardium with diminished left ventricular function up to the requirement of extracorporeal oxygenation (ECMO) is reported regularly [6]. With the ongoing pandemic due to SARS CoV-2, incidence of PIMS-TS is constantly rising, outnumbering expected cases of Kawasaki disease in very affected areas [7]. Due to the clinical and pathophysiological relatedness of these two hyperinflammatory diseases, therapy of PIMS-TS is recommended to comprise administration of immunoglobulin G and high dose acetylsalicylic acid [1]. As affected patients with PIMS-TS are usually older and therefore, have increased body weight, rheological reservations concerning hyperviscosity-associated symptoms due to high dose immunoglobulin therapy were already described [8]. Herein, we present a single-center experience of three consecutive cases of PIMS-TS that occurred over a three months period, successfully treated with serial doses of immunoglobulin G and high-dose acetylsalicylic acid, as already described elsewhere [1]. Additionally, an estimated incidence rate of PIMS-TS and SARS CoV-2 positive patients according to positive PCR tests among children and adolescents in our catchment area of 217.425 inhabitants is given.

## 2. Case Presentation

Case 1:

A previously healthy 14 year old male adolescent of mixed Turkish and German ethnicity presented at our emergency ward with headache, fever up to 39.9 °C, and vomiting since three days. Asymptomatic SARS CoV-2 RNA detection via PCR three months earlier was reported. Moreover, he suffered from decreased appetite, sore throat, and pain during swallowing. In addition, he complained about sleepiness and photophobia. On admission, he was febrile (38.6 °C) and showed tachycardia (130/min), the clinical examination revealed enlarged tonsils and deep-red inflamed pharynx, both eyes demonstrated slight, non-suppurative conjunctivitis. Pulmonary auscultation was normal, also no exanthema on the whole dermis was noted. Blood tests revealed elevated inflammatory markers paralleled by an activated coagulatory system and high cardiac injury biomarkers indicating participation of the myocardium (Table 1).

Echocardiographic evaluation showed normal heart function, coronary arteries, and valves (Figure 1).

A small pericardial effusion was revealed (Table 2) and electrocardiogram (ECG) showed no abnormalities.

The patient was started on i.v. antibiotics for presumed bacterial infection with consecutive clinical deterioration. After exclusion of acute viral and bacterial infection (Table 3), positive SARS CoV-2 IgG was retrieved, encountering asymptomatic Covid-19 earlier. Therefore, and after reviewing all clinical and laboratory data, diagnosis of PIMS-TS was made. Consecutively, an immunomodulatory therapy using immunoglobulin G (i.v.) was given, according to Kawasaki disease guidelines. As the patient’s weight was 73 kg, concerns about hyperviscosity after administration of 2 gr/kg BW in one dose were raised. Therefore, we decided to administer 2 gr/kg BW immunoglobulin G on two consecutive days (1 gr/kg BW each day) with good tolerability. The patient also received—in concordance to the Kawasaki disease treatment protocol—high dose acetylsalicylic acid. Twenty-four hours after administration of the second dose, the patient was afebrile and his clinical status improved. Acetylsalicylic acid dose was reduced to 5 mg/kg BW once daily, echocardiographic control some days later revealed resolution of the aforementioned pericardial effusion. Antibiotic therapy was ended after five days, and the patient was discharged from hospital on day 13. At follow-up two weeks later, he remained asymptomatic and in good condition.

Case 2:

A 17-year-old female adolescent of German ethnicity with prior good health conditions presented to our hospital with two days of history of itching exanthema beginning at the arms and feet, spreading secondarily over the whole body. She complained about ocular globe pain and swelling around the eyes. The patient also reported being unwell and experiencing fever chills and sweats. In addition, she complained of shortness of breath and pain in the neck. Following an uncomplicated infection of the upper respiratory tract, she was tested positive for SARS CoV-2 via PCR four weeks ago. At that time, recovery was fast, experiencing no complications. On admission, she presented in slightly decreased general condition, showing erythematous, non-scaling, itching exanthema on the whole body with predilection sites at the back, palms, and face. Auscultation of lungs and heart was unremarkable, transcutaneous oxygen saturation was 98%. Pharyngeal inspection demonstrated inflamed oral mucosa. Bilateral non-suppurative conjunctivitis was present, no enlarged lymph nodes could be detected. Neurological examination was unremarkable. Blood tests showed elevated levels for CRP, ferritin, D-Dimer, soluble Interferon 2-receptor (Table 1). Viral and bacterial detection tests, except for SARS CoV-2 antibodies, were negative (Table 3). Echocardiography demonstrated reduced left ventricular function and pericardial effusion along with elevated myocardial injury biomarkers (Table 1 and Table 2) (Figure 2 and Figure 3).

ECG resulted unremarkable. Given the combination of typical symptoms, PIMS-TS were diagnosed and consecutively a therapy with i.v. immunoglobulins G and high-dose acetylsalicylic acid was started. Due to concerns of hyperviscosity after administration of immunoglobulin G at a dose of 2 gr/kg BW in one dose, a division of immunoglobulin G dose to 1 gr/kg BW on two consecutive days was undertaken, as the girl’s weight was 76 kg. This therapy was well tolerated. Twenty-four hours after administration of the second dose, the young woman was afebrile and in good condition. Elevated inflammation markers improved after being afebrile (Table 4), echocardiographic follow-up also demonstrated improved cardiac function and diminished pericardial effusion (Table 5). The patient was discharged on day 7 with dramatically improved general condition. Echocardiographic control two weeks later showed normalization of cardiac function.

Case 3:

A previously healthy seven-year-old German girl presented to our hospital with fever up to 39.8 °C, headache, and belly pain for three days. She complained of sore throat, pain during swallowing and shortness of breath on exertion. She vomited repeatedly, oral ingestion of fluids and food was markedly reduced. Four weeks ago, she was tested positively for SARS-CoV-2, being only mildly affected with temporary loss of smell. Her parents were also tested positive at that time. On admission, the girl presented with reduced general condition, her body temperature was 39.5 °C, showing red, cracked lips, a scant erythematous rash on the back and reddened pharynx. A slight bilateral non-suppurative conjunctivitis was also noted. Her heart rate at rest counted 140 beats/min. The patient also showed elevated respiratory frequency with a transcutaneous oxygen saturation of 95%. No enlarged lymph nodes could be detected and no peripheral edema was noted. Neurological examination was unremarkable. Blood test revealed elevated inflammatory markers (Table 1) with no evidence of bacterial infection (Table 3). After four weeks of being tested positive for SARS CoV-2 by PCR, the test was still positive with a cycle threshold (ct) value of 34.8, possibly indicating not being contagious anymore (Table 3). Therefore, therapy with remdesivir was not initiated. High titer of SARS CoV-2 IgG was also detected, suggesting immunity against SARS CoV-2. Echocardiography showed reduced left ventricular function, accentuated coronary arteries close to the outlet, but no aneurysm and a scant pericardial effusion (Table 2) (Figure 4).

ECG demonstrated no abnormalities. After excluding bacterial infection and COVID -19, the diagnosis of PIMS-TS was established and immunoglobulins (2 gr/kg BW in two divided doses administered twenty-four hours apart) were given with respect to diminished left ventricular function and hyperviscosity. High-dose acetylsalicylic acid p.o. was also added. The therapy was well tolerated, twenty-four hours after the second dose, she remained afebrile and her general condition improved. After decline of fever, acetylsalicylic acid dose was reduced to 5 mg/kg BW/day. Follow-up of inflammatory and cardiac markers showed improved values (Table 4), echocardiographic assessment after therapy revealed normalized left ventricular function and coronary outlets with no pathological findings (Table 5). The patient was discharged on day 9 in good clinical condition. Follow-up some days later revealed no further complications.

Incidence of SARS CoV-2 and PIMS-TS

Table 6 summarizes the general characteristics of PIMS-TS patients.

We estimated incidence rates per year applying the Wilson method [9]. The incidence rate, for those under 18 years of age, of SARS CoV-2 positivity and of PIMS-TS among all SARS CoV-2 positive tested children for our administrative district (Ostallgäu) and our urban district (city of Kaufbeuren) are presented (Table 7).

The total number of inhabitants on 31 December 2020 in the whole region was 217.425, of which 32.697 under 18 years of age [10]. During a period of 50 weeks (week 12 in 2020 to week 8 in 2021), 570 SARS CoV-2 positive cases under 18 years tested by PCR were reported to the local health authority. During this period, three PIMS-TS patients were encountered during 15 weeks, equivalent to an incidence rate of 1.8% per year for SARS-CoV-2 positivity (95% CI: 1.7% to 2.0%) and among them 1.7% per year (95% CI: 0.9% to 3.1%) for PIMS-TS, after extrapolation to one complete year (SARS-CoV-2 positivity 593 cases, PIMS-TS cases 10 cases).

## 3. Discussion

We report a single-center experience of three children and adolescents presenting with PIMS-TS, treated with serial immunoglobulins and high-dose oral acetyl salicylic acid with good short-term outcome, in accordance with previously published reports [11,12]. A recent report stated that, in children with Kawasaki disease over 25 kg body weight, dosage of immunoglobulin could be reduced to 1 gr/kg BW without difference in outcome [13]. To date, there is no data for PIMS-TS, so we decided not to reduce the cumulative dose, but altered the administration mode by dividing in two over two consecutive days. PIMS-TS is a formerly unknown disease in children and adolescents linked to COVID-19, sharing criteria of typical or atypical Kawasaki syndrome, toxic shock syndrome, or macrophage activation syndrome [14]. The German Society of Pediatric Infectiology (DGPI) has provided a case definition of PIMS-TS, which differs from the one published by the WHO in terms of fever duration [15] (48 h of fever vs. more than 72 h of fever in the WHO’s definition). Therefore, in case of fulfillment of other criteria necessary, diagnosis might be made more often. Since the first description of this syndrome, numerous cases have been published worldwide [16,17,18], sometimes leading to harmful disease or even death. Early recognition of this entity is important since early therapy can prevent a catastrophic course of the disease [17]. The three children presented in this case series had a short pre-diagnosis period and significantly diminished left ventricular function, so therapy with immunoglobulins potentially lead to rapid recovery due to suppression of inflammatory response. So far, there is a lack of randomized controlled studies concerning treatment of this new entity and only expert opinion exists to treat PIMS-TS with immunoglobulins and salicylates [19]. All three patients were healthy, with no prior health problems reported.

In contrast to published cases, stating a majority of African-Caribbean descent, all children presented here were of European descent [20]. Echinocytes were not detected in blood swabs, neither initially nor later. This is in contrast to other reports, in which the detection of burr cells in the blood film was a hint to the diagnosis of PIMS-TS [21]. All patients in our series showed clinical signs resembling atypical Kawasaki disease, with affection of mucosa, skin, and high spiking fever, unaltered by antibiotic therapy. EBV testing in all patients revealed positive IgG values, reflecting previously endured EBV infection, as EBNA IgG antibodies also showed positive results. To date, the role of acute or endured EBV infection in the pathophysiology of PIMS-TS remains to be defined. Because cardiac function was markedly reduced in all patients and as the body weight in two adolescents equaled adult weight and because of concerns of sequelae of hyperviscosity (e.g., renal failure due to osmotic injury, hypertension), we decided to split the dose of immunoglobulins as recommended in previously published reports [22] and according to Kawasaki disease guidelines into two equal doses of 1 gr/kg BW administered every twenty-four hours. This therapy was well tolerated, no cases of hyperviscosity syndrome paralleled by worsened cardiac function were noted. As all patients recovered rapidly, other adjunctive therapies as dexamethasone or immunomodulating biologic agents, like Anakinra or Tocilizumab were not necessary [7] and ICU admission was not required.

This case series highlights the importance of early recognition of PIMS-TS in children and adolescents, as delayed diagnosis can lead to significant morbidity and mortality. These three cases treated during a three months period in winter in a relatively small pediatric ward in a rural area in Germany might give a hint to a greater number of patients affected with PIMS-TS in larger metropolitan regions. Moreover, the present case series potentially points to a greater disease burden following (often) unrecognized SARS CoV-2 infection in children (with no COVID-19 vaccine for children less than 16 years licensed yet). An incidence of 1.8% of SARS CoV-2 positive tested children in our region reflects mainly to asymptomatic children tested due to positive cases within their family or school. Accordingly, numbers might be greater among young people. Moreover, SARS CoV-2 PCR cycle threshold value should be taken into account while interpreting positive cases, as high ct values might reflect reduced or even nullified contagiosity of persons tested positive. Calculation of incidence rate of SARS CoV-2 cases and PIMS-TS cases in our region has several limitations. Due to the relatively small number of PIMS-TS, the calculated incidence of 1.7% might not reflect the true number of cases that occurred among young inhabitants of our region, as more severe cases of PIMS-TS could have been treated in larger tertiary pediatric intensive care wards in other districts nearby. Moreover, the small number of PIMS-TS cases affects the calculated incidence enormously, even only one case more during the 15 weeks of observation would increase the incidence rate for PIMS-TS in our region by one-third. Nevertheless, to the best of our knowledge, this is the first report in which a local incidence rate in a district in Germany in children with this relatively new disease was estimated.

Finally, one has to shed light on economic issues as well, since therapy with immunoglobulins is expensive [7]. As the future incidence of PIMS-TS and its long-term sequelae (e.g., development of coronary aneurysm or myocardial disease) due to the ongoing pandemic is yet unknown, this might lead to rising numbers of young patients with heart disease affecting their later social and economic life.

## 4. Conclusions

PIMS-TS in children is a serious disease following symptomatic or asymptomatic COVID-19. This case series highlights the importance of an early diagnosis and individualized therapy. With the ongoing pandemic and rising numbers of COVID-19 in children and adolescents, incidence of PIMS-TS is expected to rise.

## Figures and Tables

**Figure 1 healthcare-09-00481-f001:**
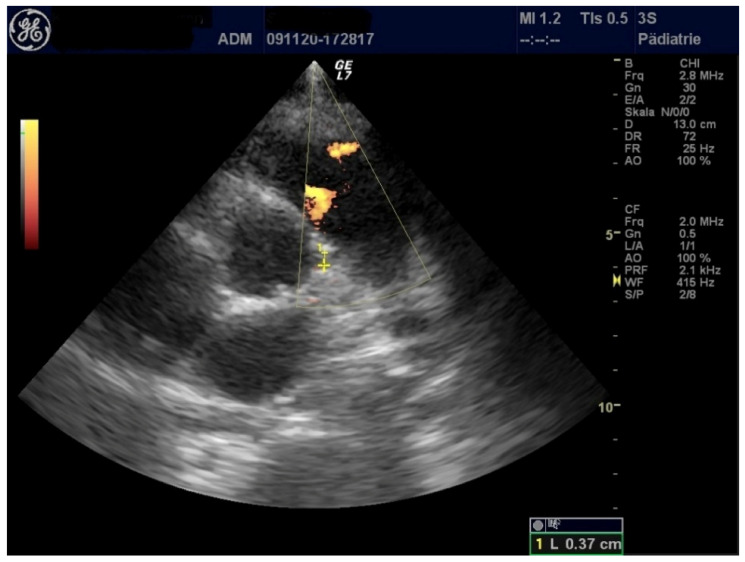
Left coronary artery with normal diameter.

**Figure 2 healthcare-09-00481-f002:**
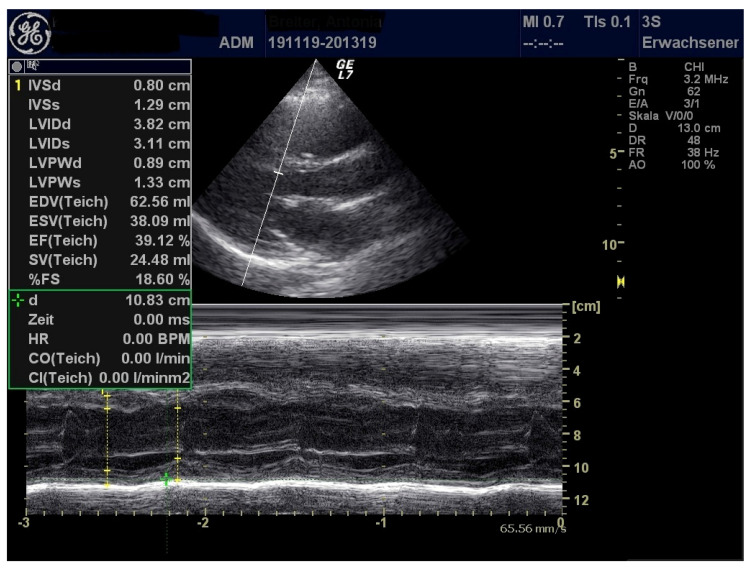
Diminished left ventricular function.

**Figure 3 healthcare-09-00481-f003:**
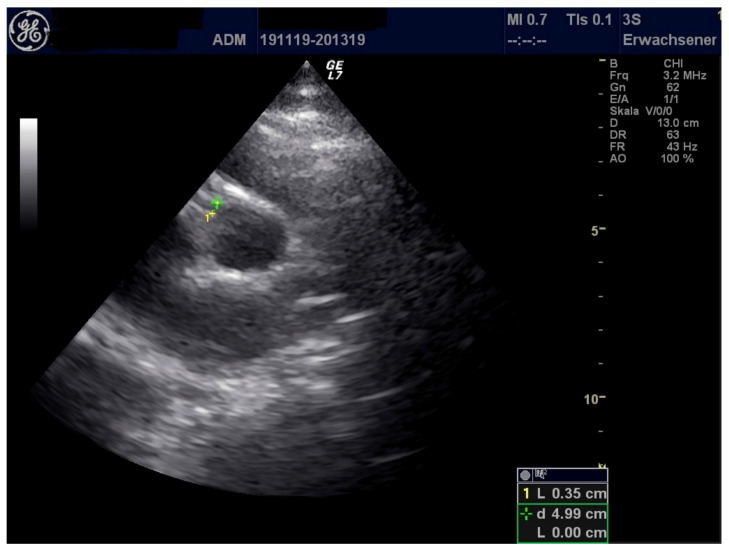
Enhanced echogenicity of right coronary artery (RCA) with normal dimension.

**Figure 4 healthcare-09-00481-f004:**
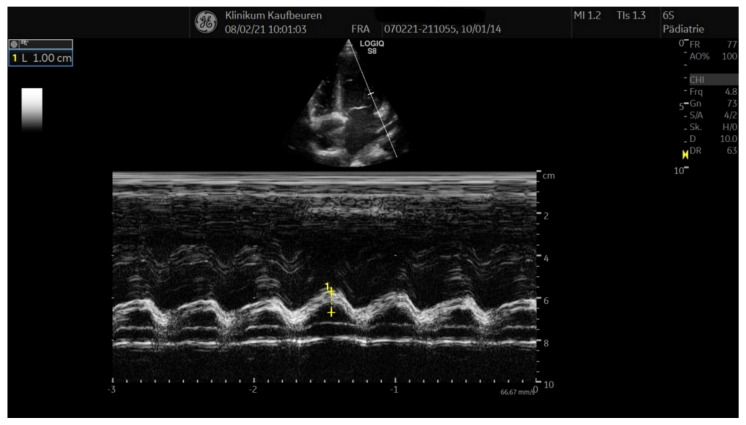
Diminished left ventricular function.

**Table 1 healthcare-09-00481-t001:** Initial laboratory findings.

	Reference Values	Patient 1	Patient 2	Patient 3
CRP (mg/L)	<5.0	257.6	119.1	135.1
PCT (ng/mL)	<0.5	0.56	0.44	1.88
Ferritin (ng/mL)	15–150	>1000	356.4	161.9
D-Dimer (µg/L)	<0.5	10.54	2.46	1.67
GOT (U/L)	<35	108	63	46
GPT (U/L)	<35	72	68	35
GGT (U/L)	<39	95	39	13
Albumin (g/L)	35–52	26.6		26.5
Troponin T (pg/mL)	<14	45.3	70.5	16.3
Leukocytes (G/L)	3.96–10.4	8.66	7.81	10.93
Neutrophile Granulocytes (G/L)	50–70%	7.15	6.59	9.54
Blood smear		No burr cells	No burr cells	No burr cells
Lymphocytes (G/L)	15–40%	4.80	7.57	8.74
sIL2Receptor (U/mL)	158–623	7253	2361	2558
NT-proBNP (pg/mL)	<125	4893	1520	1703

Abbreviations: CRP: C-reactive protein, PCT: Procalcitonin, GOT: Glutamate-oxalacetate transaminase, GPT: Glutamate-pyruvate transaminase, GGT: Gamma glutamyl transferase, sIL2 Receptor: Soluble IL2 receptor, NT-proBNP: N terminal pro brain natriuretic peptide.

**Table 2 healthcare-09-00481-t002:** Initial echocardiographic findings.

	Patient 1	Patient 2	Patient 3
LVEF (%)	71	44	53
FS (%)	40	21.1	26.6
MAPSE (cm)	1.97	1.42	0.91
TAPSE (cm)	2.55	2.08	1.14
LCA (mm)	3.8	3.3	3.1
RCA (mm)	3.2	4.0	2.8
Pericardial effusion (mm)	7	3	5
Pleural effusion (mm)	9 (left), right 0	0	0

Abbreviations: LVEF: Left ventricular ejection fraction, FS: Fractional shortening, MAPSE: Mitral annular plane systolic excursion, TAPSE: Tricuspid annular plane systolic excursion, LCA: Left coronary artery, RCA: Right coronary artery.

**Table 3 healthcare-09-00481-t003:** Serology, bacterial, and viral results.

	Patient 1	Patient 2	Patient 3
Streptococcus group A RAD test	negative	negative	negative
Adenovirus IgM	negative	negative	negative
Adenovirs DNA	negative	negative	negative
CMV IgG	negative	negative	negative
CMV IgM	negative	negative	negative
EBV VCA IgG	negative	positive	positive
EBV VCA IgM	negative	negative	negative
EBV EBNA IgG	positive	positive	positive
SARS CoV2 IgG	positive (43.6)	positive (75.3)	positive (143)
SARS CoV2 RNA nasal swab (Ct value)	negative	negative	positive (34.8)
Blood cultures	negative	negative	negative
Urin cultures	negative	negative	negative

Abbreviations: RAD test: Rapid antigen detection test, CMV: Cytomegaly virus, EBV VCA: Ebstein Barr virus viral capsid antigen, EBNA: Ebstein Barr nucleic antigen, ct: threshold value.

**Table 4 healthcare-09-00481-t004:** Laboratory findings after treatment.

	Reference Values	Patient 1	Patient 2	Patient 3
CRP (mg/L)	<5.0	33.5	40.6	135.4
PCT (ng/mL)	<0.5	3.2	n.d.	n.d.
Ferritin (ng/mL)	15–150	422.6	322.9	192.7
D-Dimer (µg/L)	<0.5	0.96	1.37	1.25
GOT (U/L)	<35	37	39	36
GPT (U/L)	<35	33	39	35
GGT (U/L)	<39	25	27	11
Troponin T (pg/mL)	< 14	21.9	7.2	19.2
Leukocytes (G/L)	3.96–10.4	7.69	4.28	11.28
Neutrophile Granulocytes (G/L)	50–70%	5.17	1.88	9.0
sIL2-Receptor (U/mL)	158–623	n.d.	n.d.	n.d.
NT-proBNP (pg/mL)	<125	49.4	503.7	80.3
	(1)			

**Table 5 healthcare-09-00481-t005:** Echocardiographic findings after treatment.

	Patient 1	Patient 2	Patient 3
LVEF (%)	81	69	73
FS (%)	50	39	41
MAPSE (cm)	n.d.	1.67	1.5
TAPSE (cm)	n.d.	2.46	2.08
LCA (mm)	3.7	3.2	2.8
RCA (mm)	3.3	3.0	2.4
Pericardial effusion (mm)	0	3–7	<3
Pleural effusion (mm)	0 (both sides)	0	0

**Table 6 healthcare-09-00481-t006:** Patient characteristics.

	Patient 1	Patient 2	Patient 3
Gender	male	female	female
Age (years)	14	17	7
Ethnicity	German-Turkish	German	German
Co-morbidities	none	none	none
Co-medication	none	none	none

**Table 7 healthcare-09-00481-t007:** Estimated incidence and confidence interval (CI).

	Rural District Ostallgäu	City District Kaufbeuren
Total inhabitants	171.226	46.199
Inhabitants under 18 years	25.247	7.450
SARS CoV-2 positive (week 12 in 2020 to week 8 in 2021)	404; female: 191	166; female: 77
PIMS-TS (1 December 2020–31 August 2021)	2 (both female)	1 (male)
	Rural District Ostallgäu + City District Kaufbeuren	
Incidence SARS-CoV-2 positive (%)	1.8%, 95% CI: 1.7% to 2.0%	
Incidence PIMS-TS (%)	1.7%, 95% CI: 0.9% to 3.1%	

## Data Availability

The data presented in this study are available on request from the corresponding author.

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
