# Peer review of "Successful Treatment of Pediatric Inflammatory Multisystem Syndrome Temporally Associated with COVID-19 (PIMS-TS) with Split Doses of Immunoglobulin G and Estimation of PIMS-TS Incidence in a County District in Southern Germany"

_healthcare, 2021, doi:10.3390/healthcare9040481_

Round 1

Reviewer 1 Report

Introduction is quite simple; this is a relevant MS and the introduction should explore the problem deeply, an initial description of PIMS is needed. The most important question here is the criteria that has conducted the selection of the 3 cases and if they are the only ones or there are others with different ending as reported in previous bibliography. 

Reviewer 2 Report

Statistical analysis should be better defined and described. Some english errors should be corrected.

Reviewer 3 Report

This is an excellent introductory report of a new entity developing as a sequelae of Covid-19.

Following corrections/clarifications are recommended:

  1. Table 3 lists the serology of various viral infections that can potentially cause similar symptoms. However, there is no mention of testing all three cases for streptococcus pharyngitis. Although strep pharyngitis may not entirely present this way, many symptoms described in these three cases overlap with a strep infection. If the information about testing for strep is present, we recommend adding it to the manuscript. If it not present, we recommend adding a sentence stating that not performing a test for strep pharyngitis may be a limitation of our study.
  2. Table 1  lists the values of various inflammatory and lab values of each patient. We recommend adding a reference range of normal values for each marker since different units are used in other parts of the world. Having a reference range can make it easier for the reader to understand how far off the listed values are from the normal range. 
  3. Description of Table 2 is missing the word "plane" when writing the full form of TAPSE.
  4. In table 3, there seems to be evidence of immunity towards EBV as evidenced by presence of IgG to EBV in all three patients . Please comment on whether this is from immunization or it indicates prior EBV infection. If its from an infection, could it be possible that this immunity status may have had some role towards development of PIMS-TS along with Covid-19 infection? It would be appropriate to include a sentence stating that further research looking into its association may be needed. 
